# Functional and evolutionary synergy of trait components can explain the existence of leaf masquerade in katydids

J. Benito Wainwright[1,2]*, Charlotte E. J. Rolfe[1], Graeme D. Ruxton[1], Nathan W. Bailey[1]

1 Centre for Biological Diversity, School of Biology, University of St Andrews, St Andrews, United Kingdom,
2 Smithsonian Tropical Research Institute, Balboa, Ancón, Panamá City, Republic of Panamá

* jbw21@st-andrews.ac.uk

## Abstract

One of the most enduring mysteries in biology concerns the evolution of complex adaptations made up of interacting component traits. When these component traits do not enhance fitness independently of one another, their origin requires that they evolve sequentially through intermediate steps that do not produce their full adaptive value as a combined trait, or alternatively, that they arise via simultaneous, synergistic evolution. We tested these alternatives using the powerful but accessible example of leaf masquerade in katydids, where in some species, highly modified wings strikingly mimic vegetation to avoid predator recognition. Combining a field predation experiment with a phylogenetic comparative analysis of wing morphology in 58 Neotropical katydid species, we show that color and shape synergistically interact to enhance survival in the wild, and modifications in both traits evolved concurrently during diversification of this clade. Our findings identify the adaptive value of masquerade camouflage in the wild and show how concordant evolutionary change in separate traits—evolutionary synergy—can generate extraordinarily specialized, multi-component adaptations.

## Introduction

Understanding the evolutionary forces that shape phenotypic complexity is a fundamental component to all aspects of organismal biology. The degree of complexity varies widely across different traits. Composite adaptations, such as vertebrate eyes and bird wings, rely on the interaction of independent component traits to acquire an emergent synergistic function [1–3]. The evolutionary origin of such extreme adaptive complexity has been a persisting question since the birth of evolutionary biology, particularly when component traits do not appear to increase fitness additively [4,5]. For the springboard prey-trapping mechanism of *Nepenthes* carnivorous pitcher plants, it has recently been demonstrated that component traits evolved independently, likely

**Data availability statement:** All raw data and analysis R code are available from Zenodo alongside the calibrated targets used in the field experiment, all lightbox images used in the online survey and mounted tegmen images: https://doi.org/10.5281/zenodo.14585161.

**Funding:** J.B.W. was supported by an 1851 Royal Commission for the Exhibition Research Fellowship, C.E.J.R. by the University of St Andrews' Rector's Scholarship, and N.W.B. by the UK Natural Environment Research Council (NE/W001616/1). The funders had no role in the study design, data collection and analyses, decision to publish, or preparation of the manuscript.

**Competing interests:** The authors have declared that no competing interests exist.

**Abbreviations:** AIC, Akaike's information criterion; BM, Brownian motion; ER, equal rates; OU, Ornstein–Uhlenbeck; PGLS, phylogenetic generalized least-squares; PPA, phylogenetic path analysis; SEMs, structural equation models; UV, ultraviolet.

while under selection for other functions, implying that complex composite adaptations can arise through intermediate evolutionary stages unrelated to their value as composite traits, and later be co-opted for novel functions [3,6]. However, this route is not inevitable for other composite traits. An alternative scenario is that composite traits evolve when directional selection acts on individual components simultaneously [4]. In this case, individual components of composite traits should show a high degree of evolutionary correlation inconsistent with sequential acquisition through nonadaptive, intermediate stages. Historically, despite the ubiquity of complex traits, a lack of suitable examples for both comparative study and experimental manipulations has made it difficult to empirically measure their emergent fitness benefits under ecologically relevant conditions and dissect evolutionary pathways leading to their evolution in natural systems.

Leaf masquerade provides an accessible and promising example in which to evaluate these alternatives. Masquerade camouflage is the resemblance of organisms to objects in the environment such as leaves, twigs, stones, and bird droppings [7–10], causing predator misclassification independently of the background upon which they're viewed [11]. Thus, masquerade provides protection without concealment. In some katydid lineages (bush crickets; Orthoptera: Tettigoniidae; Fig 1A), the tegmina (sclerotized forewings) have repeatedly evolved an extraordinary likeness to the shape and coloration of plant leaves from a nonleaf-like ancestor [14–16]. However, it is not known how the evolution of these features was coordinated to generate masquerade. It seems intuitive to suppose that a close resemblance in shape or coloration would provide little protective value from masquerade without the other. Nevertheless, katydid species appear to exhibit continuous variation in their degree of "leafiness," which provides an opportunity to evaluate the functional and evolutionary explanations of this trait using experimental and comparative techniques.

Using artificial leaf-masquerading prey and naïve, free-living avian predators, we first tested how interactions among the component traits comprising leaf masquerade affect predator perception in the wild. Next, to assess how real leaf-masquerading species have evolved, we conducted phylogenetic comparative analysis on the tegmen morphology of 58 Neotropical katydid species and integrated this with measures of human perceptions of "leafiness." Our data allow us to disentangle the evolutionary origins of leaf masquerade and shows how functional and evolutionary synergy can explain the existence of complex morphological "design" in the natural world.

## Results and discussion

### Color and shape nonadditively enhance leaf masquerade in the wild

To determine how leaf masquerade improves individual fitness under ecologically relevant conditions, we exposed wild avian predators to artificial wing targets varying in color to mimic naturally occurring green leaves, the brown bark substrate upon which they were pinned, or an unnatural blue control (Fig A in S1 Text; see Materials and methods). Doing so meant we could confirm that any survival benefits incurred by these stimuli were a result of misclassification of the targets and not due to either

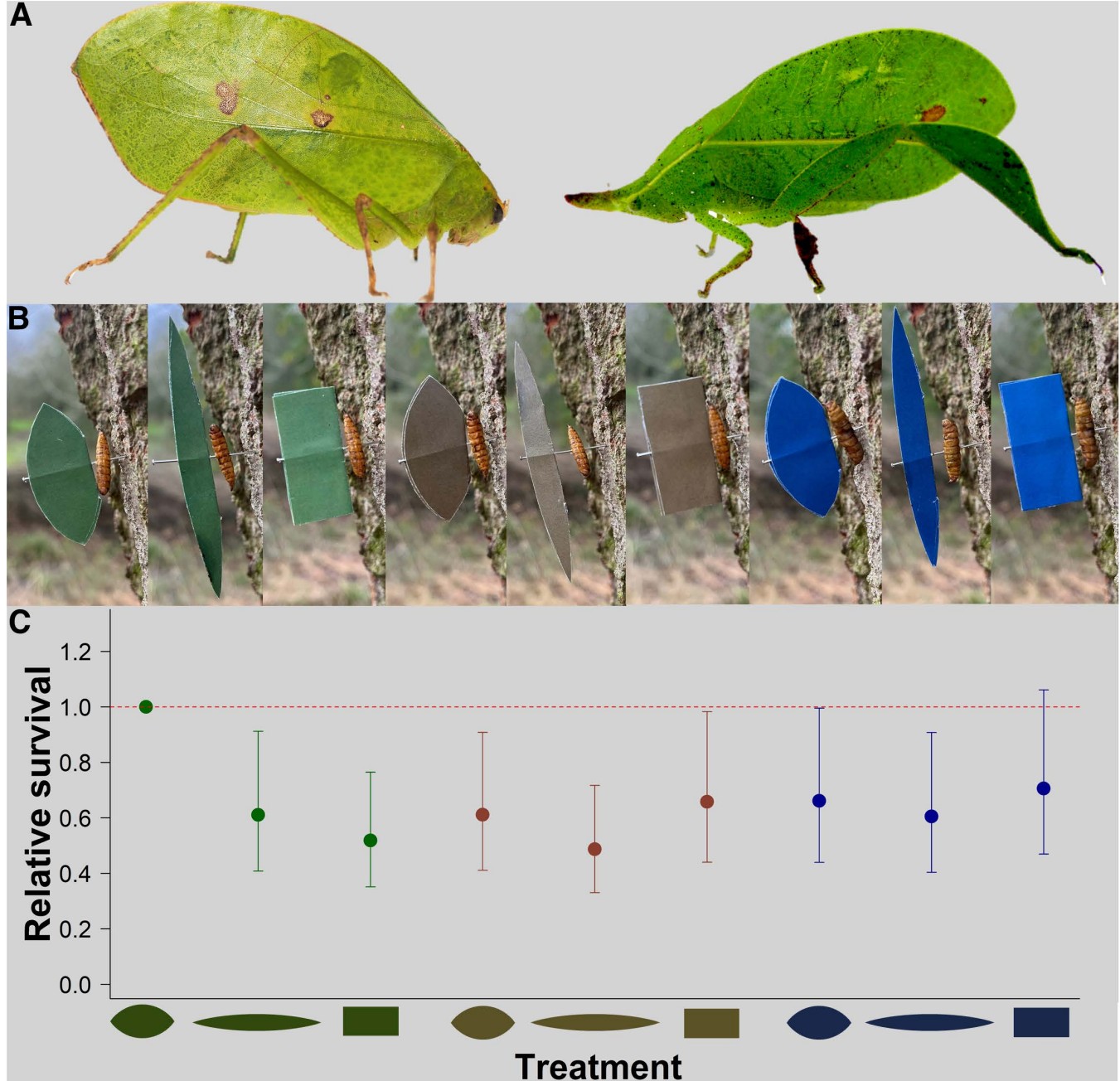

**Fig 1. Functional synergy underlies effective leaf masquerade in the wild. (A)** Photographs of Itarissa costaricensis (left) and Aegmia maculofolia (right), leaf-masquerading katydid species (Orthoptera: Tettigoniidae) found at the field site in Panama. Photos courtesy of Dr Hannah ter Hofstede and Dr Laurel Symes. **(B)** All experimental color × shape treatment combinations in situ, pinned to tree bark with a mealworm (*Tenebrio molitor* larvae) as bait. From left to right: "green oval," "green elongated oval," "green rectangle," "brown oval," "brown elongated oval," "brown rectangle," "blue oval," "blue elongated rectangle," "blue rectangle." **(C)** Odds of targets being predated, derived from a fitted Cox survival model on all color × shape treatment combinations (*N* = 1,296), relative to the "green oval" treatment, which had the highest survival rate; this model is for illustrative purposes only. Red dashed line serves as a reference for comparison, and error bars represent 95% confidence intervals. As the reference category in the model, the hazard ratio for the "green oval" treatment is fixed at 1 by definition and therefore has no associated confidence intervals [12,13]. All data underlying this figure can be found in https://doi.org/10.5281/zenodo.14585161.

enhanced crypsis of the targets against their background or to neophobic responses. Models of each color also varied in shape to mimic an oval leaf, an elongate leaf, or a rectangular control (Fig 1B), and were designed based on morphological measurements taken from the tegmina of an existing leaf-masquerading katydid species (see Materials and methods). Survival of each individual target at sampling was assessed based on the presence or absence of an edible mealworm bait, which served as the "body" of the target. To minimize potential effects of predator learning through search image formation—a phenomenon known to improve foraging efficiency in birds [17–20] —we deliberately performed the experiment in a locality where avian predators are naïve to leaf-masquerading insects. This approach, which mimics previous studies testing the adaptive value of protective coloration in tropical species using naïve predators in UK field environments [21,12], recreates the ecological conditions under which leaf masquerade first conferred a selective advantage, allowing us to isolate its functional properties, independently of learned recognition. Since masquerade relies on the implicit assumption that prey are misclassified as their models, performing this experiment in a community of "inexperienced" predators that do not routinely encounter leaf-masquerading prey enabled us to better ensure a leaf-like prey object would be classified as unprofitable by an insectivorous bird. Overall, 39.6% of targets (513) were attacked by birds (51.9% showed signs of nonavian predation, 1.9% were lost, 6.3% survived until the end of the trial). We found that the specific combination of green color and oval shape increased fitness, rather than each trait improving survival in isolation (Fig 1C and Fig B in S1 Text).

The interaction between color and shape predicted survival rate (mixed-model Cox regression: $\chi^2_4 = 9.882$, $N = 1,296$, $p = 0.043$), but neither factor was significant as a main effect (mixed-model Cox regression: color, $\chi^2_2 = 2.147$, $N = 1,296$, $p = 0.342$; shape, $\chi^2_2 = 5.878$, $N = 1,296$, $p = 0.053$). Furthermore, color affected survival rate only for oval-shaped targets (mixed-model Cox regression: oval, $\chi^2_2 = 7.143$, $N = 432$, $p = 0.028$; elongated oval, $\chi^2_2 = 2.058$, $N = 432$, $p = 0.357$; rectangle, $\chi^2_2 = 2.814$, $N = 432$, $p = 0.245$; Fig 1C and Table A in S1 Text for post-hoc tests). The "green oval" targets did not match the background color, and their shape were specifically designed to closely resemble that of an existing, but not local, leaf-masquerading katydid species, and likely other locally abundant leaves at the study site. Therefore, their increased survival compared to targets with brown, background-matching coloration and those with atypical leaf shapes strongly suggests that their survival advantage resulted from predators misclassifying them. This field experiment thus confirms an assumption central to the definition of masquerade that distinguishes it from other forms of camouflage [9,10,22].

## Composite trait evolution in Neotropical leaf-masquerading katydids

Our field predation experiment suggests that leaf masquerade is a composite trait reliant on the synergistic effects of color and shape for its adaptive function of causing predator misclassification. To assess whether color and shape evolved synergistically in real leaf-masquerading species, we studied the tegmina of 288 wild-caught individuals of 58 sympatric katydid species (196 male, 92 female; see Table B in S1 Text) from a diverse Panamanian rainforest community (Fig 2A). The coloration of each species was classified (and independently verified; see Materials and methods) as being green or brown in color and pigmented or not (Fig 2B; katydid tegmina can be translucent or opaquely pigmented irrespective of their general color, with the former being the most likely ancestral state for Tettigoniidae). Our particular interest was in tegmina that were green *and* pigmented, since green targets provided better masquerade in the avian predation experiment above. Spectrophotometry from a random subset of tegmina confirmed minimal ultraviolet (UV) reflectance, indicating that the appearance of the artificial targets used in the avian predation experiment are comparable to real katydid tegmina, in that both reflect minimal UV (Fig C in S1 Text; see Materials and methods). All studied katydid species were cryptic in coloration and available data suggests they occupy similar rainforest microhabitats [24]. Therefore, evolutionary shifts in color and pigmentation do not appear to be driven by habitat-specific background matching.

Shape was quantified by measuring the tegmen aspect ratio (maximum length/ maximum width; see Materials and methods). Thus, species with wider, more rounded, "leaf-like" tegmina had lower aspect ratios [25] (Fig 2C). The ecological relevance of our shape treatments in the avian predation experiment was supported by the observation that the

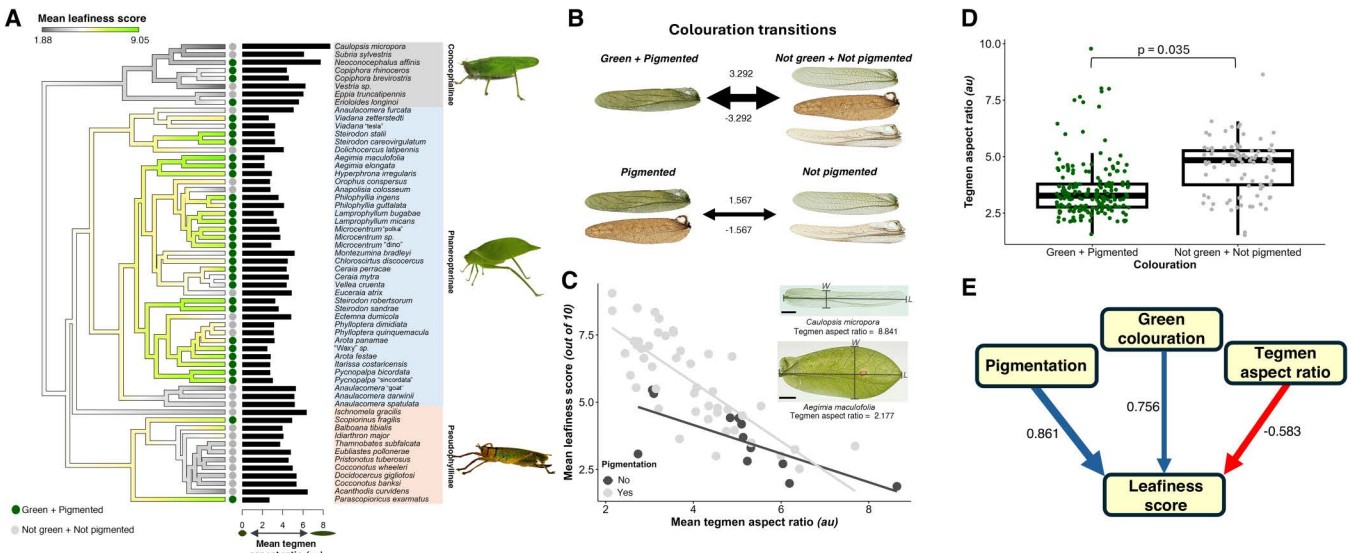

**Fig 2. Synergistic evolution explains the origins of leaf masquerade in katydids. (A)** A continuous trait plot reconstructing patterns of human perceived "leafiness" ("mean leafiness score") per species across a pruned molecular phylogeny of 58 katydid species [23]. The presence (dark green) and absence (gray) of green pigmentation is shown at the tips, and the mean tegmen aspect ratio (au) per species is indicated by black horizontal bars. Exemplars from each subfamily (highlighted blocks) are shown on the right. Photos courtesy of Ciara Kernan, Dr Laurel Symes and Dr Hannah ter Hofstede. **(B)** Transition rates between coloration states (above: green pigmentation, below: pigmentation). Arrow thickness is scaled by the transition rate value which is also given beside each arrow. **(C)** Nonadditive effects of tegmen pigmentation (presence: dark gray, absence: light gray) and mean tegmen aspect ratio on the mean leafiness score per species (*N* = 58). The inset shows mounted tegmina of the species with the lowest (*Caulopsis micropora*) and highest (*Aegimia maculofolia*) mean aspect ratio, with the measured minor (*W*) and major (*L*) axis labeled for each. Scale bars = 5 mm. **(D)** Evolutionary association between tegmen aspect ratio (*N* = 288) and the presence and absence of green tegmen pigmentation across the 58 katydid species. Thick bars indicate medians, boxes indicate interquartile ranges, and whiskers show values within 1.5 interquartile ranges. **(E)** Phylogenetic path analysis illustrating the directional and simultaneous evolutionary acquisition of pigmentation, green coloration, and reduced aspect ratio during the evolution of leaf masquerade. Human perceived "leafiness score" (out of 10) per species is the response variable. Arrows indicate the direction of the interaction, and values adjacent to these arrows are the respective path coefficients indicating the strength of the estimated relationship. Positive coefficients (blue arrows) indicate a transition from absence to presence in the parent trait (tegmen pigmentation and green coloration) whereas negative coefficients (red arrows) indicate a decrease in the parent trait (mean tegmen aspect ratio). All data underlying the graphs shown in this figure can be found in https://doi.org/10.5281/zenodo.14585161.

species with the lowest aspect ratio (*Aegimia maculofolia,* aspect ratio = 2.177) closely matched that of the 'oval' treatment in our field experiment, while the species with the highest aspect ratio (*Caulopsis micropora,* aspect ratio = 8.841) closely matched that of the 'elongated oval' treatment. Expanding upon previous categorisations of Neotropical katydids [15], two "human leafiness metrics" were obtained by asking 64 naïve human participants to both classify (presence versus absence) and score (out of 10) leafiness of the 58 species' tegmina via an online survey (see Materials and methods). Unlike previous categorisations of leaf masquerade that relied on specific morphological features [4,14], scoring "leafiness" in this way provides a holistic estimate of how leaf masquerade is perceived, without relying on any one component trait. Using a recent molecular phylogeny [23], we then reconstructed the evolution of leafiness and its shape and color component traits.

We found that the evolution of leaf masquerade is under strong phylogenetic constraint. Both green pigmentation and shape show a strong phylogenetic signal (green pigmentation, $\chi^2_1 = 0.000$, $\lambda = 1.000$, $p = 1.000$; mean aspect ratio, $\chi^2_1 = 46.132$, $\lambda = 1.042$, $p < 0.001$) and transition rates between presence and absence of green pigmentation are equivalent (Fig 2B; Fig D in S1 Text and Tables C and D in S1 Text). The most likely pattern of aspect ratio evolution followed a Brownian motion (BM) model (BM versus Ornstein–Uhlenbeck [OU]:ΔAkaike's information criterion (ΔAIC) = −2.226, BM versus early burst ΔAIC = −1.786; Fig D in S1 Text). Modeling the human leafiness metrics in the same way showed a

PLOS Biology

similarly high phylogenetic signal, but this time, an OU model was best fitting (Table D in S1 Text). These results indicate a strong degree of evolutionary inertia, potentially limiting the axes of adaptive morphological evolution among leaf-masquerading species [26]. Despite these constraints, the best-fit OU model for overall leafiness suggests that this trait has evolved under directional selection across the phylogeny. Consistent with previous work, ancestral state reconstructions support multiple independent elaborations and reductions of 'leafiness' and both its component traits across the 58 studied species [14] (Fig 2A).

## Coloration and shape have nonadditive effects on human-perceived "leafiness"

In the face of this phylogenetic constraint, we sought to test whether the functional synergy of coloration and shape, indicated by our predation experiment, is concordant with human perceptions of leafiness. While humans are not natural predators of Neotropical katydids, many insectivorous New World primates are [27–29], and behavioral experiments have suggested conserved mechanisms underlying object recognition across birds and primates [30–32]. Because tegmina were presented to human participants against a white matte background (see Materials and methods), there is no biological reason why green tegmina were more likely to be classified as leaf-like over brown tegmina. Therefore, accounting for phylogenetic effects, we tested whether human perceptions of leaf masquerade depended on the interaction between shape and the presence or absence of notable tegmen pigmentation, independent of color.

When individual participant data were analyzed using phylogenetic generalized linear mixed models, the interaction between mean aspect ratio and pigmentation predicted the human leafiness score, echoing results from the avian predation experiment (MCMCglmm: $P$-mean $= -3.840$, 95% CI $= -7.522$ to $-0.136$, $P_{MCMC} = 0.045$; Fig 2A and 2C and Table E in S1 Text). When species were subset by pigmentation status, participants consistently awarded high leafiness scores to low aspect ratio tegmina, but the strength of this association was stronger for pigmented species (MCMCglmm: pigmented, $P$-mean $= -8.930$, 95% CI $= -10.495$ to $-7.234$, $P_{MCMC} < 0.001$; not pigmented, $P$-mean $= -6.144$, 95% CI $= -11.705 – -9.959$, $P_{MCMC} = 0.025$). These results were confirmed by equivalent models constructed using phylogenetic generalized least-squares (PGLS; see Table F in S1 Text). No interaction between pigmentation and shape was found when human-perceived leaf masquerade presence/absence was used as the dependent variable rather than "leafiness" score, possibly because the former metric was too coarse to capture continuous variation in leafiness (MCMCglmm: $P$-mean $= -4.772$, 95% CI $= -10.805$ to $1.531$, $P_{MCMC} = 0.120$; Tables E and F in S1 Text). While many New World primate predators of katydids are also trichromats, thus enhancing the ecological relevance of our human-based assessments of leafiness, these metrics might represent a conservative estimate of how color and shape interact in the eyes of dichromatic species, whose visual systems are thought to improve the ability to find camouflaged prey [29,33,34]. Overall, in concordance with the conclusions derived from our artificial prey experiment, perception of leaf masquerade appears reliant on the co-occurrence of appropriate coloration, shape, and possibly other independent components, for exploiting both avian and mammalian recognition systems.

## Coloration and shape co-evolve synergistically to produce spectacular leaf masquerade

Given that the interaction between coloration and shape enhances the efficacy of leaf masquerade, we next tested whether nonadaptive stepwise or simultaneous acquisition of these trait components explain its evolution. In other words, were coloration and shape selected independently at different timepoints, one maintained by selection for functions unrelated to leaf masquerade and later co-opted to form a composite masquerade trait with the other? Or did they both evolve simultaneously and result in this form of camouflage? Nonadaptive stepwise evolution would imply no evolutionary correlation between coloration and shape across the phylogeny—that is, these traits would not tend to arise at the same time during the evolutionary history of these species because neither would enhance masquerade in isolation. The opposite is predicted if these traits evolved simultaneously—that is, they should tend to arise concurrently during the species' evolutionary history. After controlling for phylogeny, MCMCglmm analysis on individual data and a phylogenetic ANOVA

on species means confirmed that the evolution of green pigmentation was significantly associated with reduced tegmen aspect ratios (MCMCglmm: $P$-mean = 0.062, 95% CI = 0.004 to 0.117, $P_{MCMC}$ = 0.035; phylogenetic ANOVA: $F_1$ = 16.127, $p$ = 0.025; Fig 2D). These results are consistent with an evolutionary scenario whereby coloration and shape simultaneously evolved under directional selection. This model also revealed that female tegmina exhibit significantly lower aspect ratios compared to males ($P$-mean = −0.014, 95% CI = −0.023 to −0.005, $P_{MCMC}$ = 0.002), suggesting that males and females likely experience different selection pressures, but the interaction between green pigmentation and sex was not significant ($P$-mean = −0.002, 95% CI = −0.016 to 0.012, $P_{MCMC}$ = 0.847).

Given the evolutionary correlation between trait components, we next assessed the order of pigmentation, green coloration, and aspect ratio acquisition across the phylogeny using phylogenetic path analysis (PPA). This method involved constructing structural equation models (SEMs) to represent alternative evolutionary sequences by which these traits were acquired stepwise across the phylogeny. We also modeled a scenario in which all three traits were acquired concurrently, as well as hybrid models that combined elements of simultaneous and stepwise acquisition of the three traits. When both leaf masquerade probability and mean leafiness score were used as the dependent variable, the path analysis yielded strong support for models of simultaneous acquisition and loss of all three traits, supporting their coordinated, synergistic evolution during the origin of leaf masquerade (Fig 2E; Fig E in S1 Text; Table G in S1 Text). Taken together, the evolution of leaf masquerade in our studied species is best explained by the simultaneous, synergistic acquisition and loss of trait components, under directional selection.

## Conclusions

Functional synergy between color and shape of leaf-like prey causes misclassification by naïve predators, and phylogenetic reconstruction of these component traits supports their simultaneous, synergistic evolution in Neotropical katydid tegmina. Our field experiment demonstrates that resemblance in both color and shape is necessary for this type of masquerade to be functional [35]. While we isolate and demonstrate the functional benefits of masquerade in a natural context and illustrate the synergistic evolution of its component traits, other sources of selection are likely to act upon the tegmina of Neotropical katydids, including sexual selection, as male tegmina are used to produce sound, and other forms of ecological selection. The latter might explain the sex differences in tegmen shape that we observed. In nature, masquerade likely also co-evolves with behavioral adaptations such as resting orientation and microhabitat selection, as has been observed in other masquerading organisms [8,36–38]. Although selection for background matching may have contributed towards the evolution and maintenance of leaf masquerade, disentangling relative strengths of these effects, particularly with little ecological data, is challenging. It is irrelevant to our arguments how much leaf masqueraders benefit from crypsis through background matching; their often-uncanny specific resemblance to leaves suggests an additional benefit through being misidentified.

By integrating experimental evidence with comparative analyses of extant species, we demonstrate how composite adaptations can originate through the simultaneous recruitment of individual trait components, rather than by stepwise evolution. Our findings validate and expand upon patterns of contingent wing pattern evolution observed in leaf-masquerading butterflies [39,40], providing a plausible evolutionary mechanism for explaining the origin of integrated phenotypes, leaving its genetic basis an alluring topic of future study. In many species, leaf masquerade appears to have been further optimized through the evolution of elaborate venation patterning, false necrotic spots, and false holes [15,21,41,42]. The sequence and pattern by which these more sophisticated leaf-specific traits are acquired across species, and their relative fitness advantages, warrants further investigation. The adaptive significance and evolutionary mechanisms underlying within-species polymorphisms remain an important avenue for further targeted study [15]. Color polymorphisms in particular are common in many of our sampled species when comparing populations across their geographic range, but here we used species for which intraspecific polymorphism was absent at our sampling locality, enabling species-level phylogenetic comparative analyses. Although not part of our analysis, genera within the subfamily

Pterochrozinae such as *Mimetica* and *Typophyllum* are exceptional examples of leaf masquerade, and these do exhibit striking within-species differences in both coloration and shape, even within the same population [15,42]. It is not known whether this variation is heritable or the result of environmentally induced plasticity, However, such extreme phenotypic lability might hint that color and shape have become evolutionarily decoupled in this lineage. Testing this will require large within-species sample sizes. It would also be interesting to test if predator learning shapes the relative fitness of these morphs among communities of experienced predators.

To conclude, the appearance of exquisitely sophisticated 'design' in the natural world is an enduring issue in evolutionary biology that propels both historical and contemporary debates [4,5,43]. While the dominant, gradualist paradigm envisions sequential optimization of adaptive traits that eventually build complex functions, we suggest that alternative scenarios warrant more serious investigation; that is, those involving the co-occurrence of traits with synergistic effects, which may be rarer, but subject to stronger selection.

## Materials and methods

### Wild katydid sampling

Adult katydids were sampled at night from lights around research station buildings on Barro Colorado Island (BCI), Panama (9.1647° N, 79.8367° W) in March–April 2024 and January–May 2025 between 04:00 and 06:00 and 20:00 and 01:00. Individuals were identified to species level using public resources [44,45] and custom ID keys provided by Dr Laurel Symes and Dr Hannah ter Hofstede. A total of 288 individuals were sampled across 58 species (51 species were sampled in 2024, with a further 7 species sampled in 2025; Fig B in S1 Text), with representatives from subfamilies Conocephalinae, Phaneropterinae and Pseudophyllinae (Figs 1A and 2A). We collected more than three individuals of 31 species. Studying a single diverse community of katydids with seemingly generalist habitat preferences [24] allowed for robust comparative analyses by eliminating variation in background vegetation composition, which might influence the evolution of different leaf masquerading strategies.

Captured individuals were placed in a freezer at −20°C for 10 min, after which morphological data were collected (body length, femur length/width, thorax height). Whole tegmina were removed with microscissors. The external surface of the tegmina was photographed using a Nikon D3300 camera (Nikon Corporation, Tokyo, Japan) with a Sigma 17–50 mm *f*/2.8 lens (Sigma Corporation, Kanagawa, Japan) in a photography light box (DUCLUS, model DU5032U, Guangdong, China) under LED illumination. Images were taken against a matte white background, at a constant distance (~25 cm) and magnification, and contained a ColorChecker passport (X-Rite Grand Rapids, MI, USA) for future coloration analysis. Tegmina were preserved as dry specimens, and the remaining body tissue was either preserved in 95% ethanol or contributed to the Museo de Invertebrados Fairchild de la Universidad de Panamá (MIUP). Samples were exported to the United Kingdom under export permit collection nos. PA-01-ARG-040-2024 and PA-01-ARB-039-2025, obtained from the Ministerio de Ambiente, Panama.

### Field experiment

**Target creation.** To test whether combinations of color and shape provide fitness benefits through masquerade in nature, artificial leaf-like stimuli were exposed to wild avian predators. Prey targets consisted of colored paper "leaves" attached to an edible mealworm "body" (*Tenebrio molitor* larvae, frozen at −80°C then thawed). All targets used mealworm baits, eliminating the potential for experimental confounds across treatments. All treatments had a total surface area of 4.85 cm$^2$ which falls within the range of (a) suitable prey typically consumed by local passerine birds and (b) leaf area variation observed in real bramble (*Rubus spp.*) leaves which these targets were designed to, at least somewhat, imitate [46]. Three treatment groups for both color and shape were created, following a 3 × 3 factorial design. Our aim was to create generic, ecologically relevant targets that could be perceived as leaf-like by wild passerine predators, without resembling the morphology of any specific insect prey species.

Because leaves (and tree bark, see below) reflect little to no UV light [47,48], standard digital photography was used to create ecologically relevant target stimuli following established approaches [e.g., 21,47,13,49–54]. To create calibrated green stimuli, digital photographs of 34 individual bramble leaves were taken using a Nikon D3300 camera (Nikon Corporation, Tokyo, Japan) with a Sigma 17–50 mm *f*/2.8 lens (Sigma Corporation, Kanagawa, Japan) under natural daylight illumination, each containing a ColorChecker passport (X-Rite Grand Rapids, MI, USA) for color calibration [49]. Bramble was chosen because it was common across our field sites, meaning local birds would have encountered its leaves, but crucially, it and any other green foliage was never found on the tree bark substrates on which targets were pinned. Images were saved in raw (NEF) format and imported into ImageJ [55] for subsequent linearization and calibration within the MICA toolbox, a widely used software attachment for conducting image analysis from the perspective of different animals' visual systems [52,55]. Multispectral images were converted to blue tit (*Cyanistes caeruleus*) visual color space, a model insectivorous passerine bird present at our field site [56]. From these cone-catch images, the mean RGB values of each leaf were obtained and averaged across all images to estimate the average color of bramble leaves as perceived by ecologically relevant predators.

To separate the effects of masquerade from those of background matching camouflage, brown stimuli that matched the average color of the substrate upon which targets were pinned formed a second color treatment group. A total of 57 photographs of oak (*Quercus robur*), ash (*Fraxinus excelsior*), sycamore (*Acer pseudoplatanus*), and field maple (*Acer campestre*) bark were taken and calibrated using the same methodology as above. The reflectance of our green and brown targets therefore fell within variation observed in the leaves and bark they were designed to resemble. As a control, blue targets that did not match any aspect of the environment were created by modifying the RGB values of the green stimulus such that the overall luminance, measured in double cone catch quanta (the avian achromatic channel [52,56]), was the same. To confirm that the addition of the brown mealworm bait would not meaningfully affect the relative contrast of the different treatments against their background, a set of recently thawed mealworms was photographed under the same lighting conditions and calibrated in the same way. As expected, the mean reflectance values for the mealworms fell within the RGB distribution of the tree bark background upon which targets were pinned (Fig A in S1 Text).

The shape of the artificial stimuli was designed based on the mean aspect ratio of tegmina estimated from *Aegimia maculofolia* (Tettigoniidae: Phaneropterinae) (2.101; $n = 7$), a highly specialized leaf-masquerading katydid species [57]. Aspect ratio is an accurate proxy of leafiness where individuals with low aspect ratios have more rounded, "leaf-like" tegmina [14,25]. The maximum length (the "major axis") and width (the "minor axis," perpendicular to the length) of the left and right tegmen was measured from JPEG light box images (see above) in ImageJ [55], averaged, and used to calculate the "tegmen aspect ratio" for this species (aspect ratio = major axis length/minor axis length). The objective was to generate a shape that was recognizably leaf-like but was not under any developmental constraint, such that it could plausibly evolve in the wings of real insect prey. A generic leaf-like shape with this aspect ratio (3.72 × 1.77 cm) was made by creating a vesica piscis using the shape tool in Powerpoint (Microsoft Corporation). An elongated treatment was created by doubling the length of the original vesica piscis while keeping the total surface area the same. The resulting shape (7.45 × 0.88 cm) had an aspect ratio of 8.47. This aspect ratio of the elongated treatment did not resemble that of most natural leaves, allowing us to control for intrinsic Gestalt properties of being leaf shaped which could predict the detectability of prey during visual search, rather than providing a benefit through misclassification (i.e., masquerade) per se. In addition, this treatment evaluated whether elongate-bodied insect species (including all sampled katydids) generally benefit from masquerade to some degree. As a control, a nonleaf shaped rectangular treatment was created, with surface area kept constant (3.19 × 1.52 cm).

Pairs of adjacent targets of the same treatment group were printed double-sided onto A4 waterproof paper (Rite-in-the-Rain, JL Darling, Tacoma, WA, USA) using a calibrated Xerox VersaLink C7030 printer (Xerox, CT, USA). These were then folded along the midline (where the boundaries of the adjacent pairs overlapped slightly), with a steel sewing pin (Korbond Industries Lincolnshire, UK) driven through the center, and glued together (UHU, Bühl, Germany).

**Protocol.** The field experiment took place from July to September 2024 and was conducted along the Lade Braes public footpath, St Andrews, Fife, UK (56.3371° N, 2.8036° W), a mixed-deciduous forested habitat, covering an area of ~1.7 km², and inhabited by a variety of insectivorous passerine birds. Replicate blocks of targets were pinned at three self-contained, unfragmented sites along this footpath, which ran parallel to the Kinness Burn (Cockshaugh Park site) and the Cairnsmill Burn (eastern and western sites). Conducting this experiment in a locality where avian predators do not routinely encounter leaf-masquerading katydids overcomes potentially confounding effects of prior experience, which could otherwise enhance predation on leaf-like stimuli (e.g., "green ovals") if predators had learned they were profitable (see main text for details). Although our UK field site contains different passerine bird species to those found in Panama where katydids were sampled, conserved mechanisms of color perception and spatial resolution across passerines mean that it is unlikely our targets would be perceived any differently by these species [58–60]. The experiment was approved by the School of Biology Ethics Committee, University of St Andrews (reference number BL17958).

The experimental procedure followed that of several preceding studies [21,12,47,13,50,51]. Targets were haphazardly selected from a plastic bag in which all targets for a block had been thoroughly mixed. These were then pinned to the bark of oak, ash, sycamore, and field maple exclusively (to ensure brown targets matched the color of their background) with a thawed mealworm threaded onto the pin (Fig 1B). Targets were pinned to the bark of tree trunks free of lichen, moss, or vegetation, at a height of approximately 1.8 m, and trees with a circumference smaller than 0.5 m were ignored. A total of 1,296 individual targets were put out across 16 experimental blocks of 81 models (nine replicates per treatment per block). Each block was conducted at a different location within each site to minimize the probability of the targets being encountered by the same individual predators. Fresh targets were made for each block.

Checks were made at 24, 48, and 72 h intervals, where the "survival" of an individual target at each check was determined by the presence or absence of the mealworm bait, with the target still present, intact, and attached to the tree. Targets were classified as "censored" if they survived until the final 72-h check, showed signs of nonavian predation (e.g., hollow exoskeleton from spiders or slime trails from slugs), were lost, or were relocated but no longer attached to the tree. In survival analysis, censored data are incomplete observations where the exact time of an event (i.e., predation) is unknown, but it is known that the target survived until a certain point. These data allow us to incorporate all available information—from both complete and incomplete records of predation events—into analyses. Predated targets were removed at each check, and all remaining targets were collected at the final 72-h check.

**Survival analysis.** Survival analysis was performed by constructing mixed effects Cox regressions with the coxme() function from the *coxme* package in R [61,62]. Target color, shape, and their interaction were treated as fixed effects, with experimental block and site included as random effects. Visual inspection of the partial residuals against the ranked survival time confirmed the proportional hazard assumption of the models. Analyses of deviance, comparing models with and without the factor of interest, were then performed using the anova() function and tested against a $\chi^2$ distribution. Subsequent post-hoc pairwise contrasts were made by creating custom contrasts using the glht() function from the *multcomp* package [63]. *P*-values were unadjusted if the number of pairwise tests was not greater than the degrees of freedom [64].

## Comparative morphology

**Katydid tegmen traits.** From the field-collected katydid tegmen samples, color and shape data were obtained to investigate how these traits coevolve to promote the elaboration of leaf-masquerading phenotypes across species. After randomly sampling a single light box image per species, a researcher scored whether the majority of the tegmen surface was (a) pigmented or not and (b) green or brown in color (Fig 2B). In "unpigmented" species, the cells created by the reticulations of the tegmen venation were clearly visible, indicating a lack of pigment in these regions, as they appeared either translucent or completely transparent. All individuals of the sampled species were monomorphic for the categorical coloration states we obtained, meaning that color classifications based on a single individual were representative of the

species as a whole at our sampling locality. Both coloration classifications were confirmed independently by a naïve researcher who scored the same images and had not been informed about the purpose of the study. Based on these classifications, species were then grouped based on whether they were green *and* pigmented or not. To confirm that katydid tegmina show minimal reflectance outside the visible range, reflectance was also measured from a random subset of eight unmanipulated katydid tegmina (four species) by mounting specimens adjacent to an integrating sphere and scanning with a V-770 spectrophotometer (JASCO, Tokyo, Japan) (wavelength range: 290–700 nm, resolution: 1 nm), calibrated using a white reflectance standard.

To ensure accurate measures of shape, a different set of images were taken using flattened tegmen samples. Dried tegmina were rehydrated in an insect relaxing chamber: a hermetic plastic box filled with paper towels soaked in water, with a small quantity of ethanol to prevent contamination, for 24 h. This procedure softened the cuticle, easing manipulation of the sample by reducing the risk of shattering. Individual tegmina were then mounted between two clean microscope slides and sealed with scotch tape. Whole specimens were photographed, accompanied by a suitable scale, with the tripod-mounted 12MP camera of an iPhone 13 Mini (Apple, Cupertino, CA, USA) under ventral LED illumination. JPEG images were imported into ImageJ [55] where the tegmen aspect ratio for each individual was calculated (Fig 2C). The length of the minor axis was measured excluding the stridulatory area which naturally forms a perpendicular fold that rests on the dorsal surface of the animal. Mean aspect ratio for each species was calculated (Fig B in S1 Text), and both individual-level and mean values were $\log_{10}$ transformed prior to analysis.

To explore how color and shape influence human perceptions of leafiness, an online survey was designed and distributed using Qualtrics (Qualtrics Provo UT, USA) in October–November 2024 where a randomly selected light box JPEG image (nonlinearized, with a ColorChecker passport included) of each of the 51 katydid species collected in 2024 was shown to 53 participants naïve to the purpose of the study (53% female, aged 18+, all with trichromatic color vision). In June 2025, an updated version of the survey incorporating an additional seven species sampled in 2025 was completed by a further 11 participants (55% female, aged 18+, all with trichromatic color vision). All participants gave their informed consent in line with the Declaration of Helsinki. For each participant, images were presented in a randomized order. Participants were given one minute per image to (a) state whether each species was "leaf-like" or not and (b) provide a "leafiness score" of each species on a 0–10 scale, where 0 is not leaf-like at all and 10 is extremely leaf-like. Once a response was submitted for each species, participants were not able to change their mind. These holistic measures of "leafiness" allowed us to study whether variation in color and shape interact to influence human perception of leaf masquerade. Our use of humans as proxies for understanding how more ecologically relevant primates detect camouflaged items has been validated by an extensive body of prior work [12,13,65–67]. The survey was approved by the School of Biology Ethics Committee, University of St Andrews (reference number BL18128).

**Phylogenetic comparative analysis.**  Using a recent phylogenetic tree for Tettigoniidae from Kernan *and colleagues* [23], pruned to include only the 58 sampled species, the phylogenetic signal of each trait was estimated using Pagel's λ [68]. To model the evolution of binary coloration traits, the fitDiscrete() function from the *geiger* package was implemented [69]. The equal rates (ER) model (where the probability of transitioning into each character state is equal) was compared with the all-rates-different model (where the probability of transitioning into each character state is unequal) based on the difference in ΔAIC. Models with a ΔAIC < 2 are considered equivalent and therefore, the model with the fewest parameters (ER) is selected. The best-fitting model was used to run the make.simmap() function in *phytools* [70] which applies a maximum likelihood approach to simulate 10,000 character maps of the phylogeny and reconstructs the evolution of coloration across the tree (Fig D in S1 Text). This provides an estimate of the coloration state at each node, including the ancestral character state.

Aspect ratio was modeled by constructing BM (random-walk evolutionary processes with no selection component), OU (stochastic variation with a deterministic selection component), and early-burst (EB; rapid diversification early in cladogenesis followed by BM) models of trait evolution using the fitContinuous() function in *geiger* [69]. As with coloration, models were compared based on ΔAIC and visualized using the contMap() function [70] (Fig D in S1 Text).

To test if coloration, mean aspect ratio, and their interaction influence human perceptions of leafiness, Bayesian phylogenetic generalized linear mixed models were constructed using the *MCMCglmm* package in R, which incorporates the inverse correlation matrix of the phylogeny as a random effect [71]. Default priors were used as fixed effects. For normally distributed continuous (family = "gaussian") response variables, uninformative, parameter expanded priors were used as random effects (G: V = 1, nu = 1, alpha.mu = 0, alpha.V = 1,000; R: V = 1, nu = 0.002), whereas with binary (family = "categorical") and ordinal response variables (i.e., Leafiness score; family = "ordinal"), weakly informative inverse-Wishart priors were used with a fixed residual variance (binary variables, G: V = 1, nu = 1, alpha.mu = 0, alpha.V = 1,000; R: V = 1, nu = 0.002, fix = 1; ordinal variables, G: V = 1, nu = 0.002; R: V = 1, fix = 1). Species, participant, and survey version were included as additional random effects and each model was run for 5,100,00 iterations, with a burnin of 100,000. A separate model also confirmed that participant responses did not significantly differ between the two versions of the survey conducted in 2024 and 2025 (leaf masquerade presence: *P*-mean = −0.699, 95% CI = −1.899 to 0.416, $P_{MCMC}$ = 0.238; leafiness score: *P*-mean = −0.182, 95% CI = −0.950 to 0.561, $P_{MCMC}$ = 0.651). Additional models were also run to test for evolutionary associations between individual-level aspect ratio and coloration. In these models, sex and its interaction with coloration were included as additional fixed effects and species was included as a random effect to control for between-species variation. In all cases, the significance of each predictor is reported as the probability of the parameter value being different from zero ($P_{MCMC}$). We also report the posterior mean (*P*-mean) and 95% credible interval for each predictor.

By calculating the proportion of participants categorizing each species as leaf-like and the mean leafiness score per species, equivalent models to those made with MCMCglmm were recreated using PGLS regression in the *caper* package [72]. Visual inspection of the residuals from the fitted models, followed by Shapiro–Wilk tests, confirmed they were normally distributed with no heteroscedasticity. Pagel's $\lambda$ for each model was estimated based on maximum likelihood. To test for an association between mean aspect ratio and coloration, phylogenetic ANOVAs were performed, under 1,000 simulations, using the phylANOVA() function in the *phytools* package [70].

Finally, we modeled support for hypothetical causal relationships between traits using PPA with the *phylopath* package [73,74]. This method integrates categorical (coloration) and continuous (mean tegmen aspect ratio) trait data to assess the interrelation between traits that result in "better" leaf masquerade, based on the human-informed metrics of leafiness described above. By building SEMs that suggest multiple scenarios by which coloration and shape are acquired during leaf masquerade evolution (Fig E in S1 Text), the best fitting model structure is indicated by the model with the lowest C-statistics information criterion. A best-fitting model structure whereby traits are acquired sequentially would therefore suggest stepwise evolution, whereas one where all traits are acquired concurrently would suggest simultaneous, synergistic evolution.

## Supporting information

**S1 Text. Supplementary information. Fig A.** Distributions of bird vision-calibrated normalized RGB reflectance values obtained from raw multispectral images of bramble leaves (top) and tree bark (bottom), which informed the color of the targets used in the field predation experiment. Vertical dashed blue lines in the bottom row indicate the mean RGB values of a random sample of recently thawed mealworms (*Tenebrio molitor* larvae), which were used as bait in the experiment and fall within the RGB distribution of the tree bark—the substrate upon which the targets were pinned. All data underlying this figure can be found in https://doi.org/10.5281/zenodo.14585161. **Fig B.** Survival plot for each color × shape treatment combination over time when exposed to wild avian predation (*N* = 1,296). Shading represents 95% confidence intervals for each treatment. All data underlying this figure can be found in https://doi.org/10.5281/zenodo.14585161. **Fig C.** Mean reflectance values for the tegmina of four randomly katydid species within a 290–700 nm spectral range (*n* = 2 per species). The inset plots these data in tetrachromatic avian color space using the blue tit (*Cyanistes caeruleus*) visual model, where points represent the photon catch of the blue tit shortwave (S), mediumwave (M), longwave (L), and ultraviolet (UV)

cones. All data underlying this figure can be found in https://doi.org/10.5281/zenodo.14585161. **Fig D.** Estimated transitions from absence (gray nodes and tips) to presence (green nodes and tips) of green tegmen pigmentation, and vice versa (left), and patterns of mean tegmen aspect ratio (au) evolution (right) across a pruned molecular phylogeny of 58 sampled katydid species [23]. Pies at each node represent the posterior probability of the ancestor having each character state. All data underlying this figure can be found in https://doi.org/10.5281/zenodo.14585161. **Fig E.** All models included in each set of the phylogenetic path analysis with mean leafiness score as the response variable. The C-statistics information criterion (CIC) is indicated below each model. **(A)** Model set included the presence or absence of green pigmentation (Green_pigmented) and mean tegmen aspect ratio (AR) as predictor variables. **(B)** Model set included mean tegmen aspect ratio and the presence or absence of pigmentation, and green coloration ("Color") as individual predictor variables. All data underlying this figure can be found in https://doi.org/10.5281/zenodo.14585161. **Table A.** Results from post-hoc custom pairwise comparisons testing the effect of color on the survival of oval-shaped targets in the wild avian predation experiment, the only shape treatment where color had a significant effect. The survival rate of the "green" treatment was compared with that of the 'brown' and 'blue' color treatments. Significant differences are denoted by asterisks. $*p < 0.05$, $**p < 0.01$, $***p < 0.001$. **Table B.** Summary data for all 58 sampled species on Barro Colorado Island, Panama ($N = 288$). **Table C.** Transition rates between coloration character states across the katydid phylogeny. Included are the log likelihood and AIC values of the equal rates (null) model and the all-rates-different model. Output from log likelihood ratio tests, comparing the two models, are also shown where a significant difference ($p < 0.05$) confirms that the all-rates-different model is accepted. **Table D.** Evolutionary modeling of continuous traits across the katydid phylogeny. Included are log likelihood and AIC values for Brownian motion (BM), Ornstein–Uhlenbeck (OU), and early-burst (EB) models. Output from likelihood ratio tests, comparing each model with the one with the lowest AIC score are also shown. **Table E.** Results from MCMCglmms which test the effect of coloration (green pigmented, pigmented), shape (mean tegmen aspect ratio), and their interaction on individual human leafiness perception (Leaf masquerade presence and Leafiness score). Shown are the posterior means, 95% credible intervals, and the effective sample size for each independent variable in the model. Significant effects are denoted by asterisks. $*P_{MCMC} < 0.05$, $**P_{MCMC} < 0.01$, $***P_{MCMC} < 0.001$. **Table F.** Results from phylogenetic generalized least-squares (PGLS) models which test the effect of coloration (green pigmented, pigmented), shape (mean tegmen aspect ratio) and their interaction on the species means of two metrics of human leafiness perception (Leaf masquerade probability and Mean leafiness score). Phylogenetic signal of each model was calculated using the maximum likelihood estimate of Pagel's $\lambda$. Significant effects are denoted by asterisks. $*p < 0.05$, $**p < 0.01$, $***p < 0.001$. **Table G.** Path coefficients from the best fitting model from four separate phylogenetic path analysis model sets, alongside the standard error and 95% confidence intervals.
(DOCX)

## Acknowledgments

We are grateful to Gregg Cohen and Rachel Page at the Smithsonian Tropical Research Institute, all staff at the Barro Colorado Island research station, Lil Marie Camacho, and Panama's Ministerio de Ambiente for logistical fieldwork support and Laurel Symes for help with katydid identification. Many thanks also to Innes Cuthill and Jolyon Troscianko for providing advice on color calibration, Rivers Tarr and Sophie Rolfe for helping with target creation, Mirabella Funk for assistance in the field, and Rebecca Meehan and George Dwapanyin for spectrophotometry assistance. Additionally, we thank Tony Robillard for providing permission to use his phylogenetic tree, Leeban Yusuf for help with coloration categorization, and all participants involved in the online survey; they were not financially remunerated.

## Author contributions

**Conceptualization:** J. Benito Wainwright, Charlotte E. J. Rolfe.

**Data curation:** J. Benito Wainwright.

**Formal analysis:** J. Benito Wainwright.

**Funding acquisition:** J. Benito Wainwright, Charlotte E. J. Rolfe, Nathan W. Bailey.

**Investigation:** J. Benito Wainwright, Charlotte E. J. Rolfe.

**Methodology:** J. Benito Wainwright, Graeme D. Ruxton, Nathan W. Bailey.

**Project administration:** J. Benito Wainwright.

**Resources:** J. Benito Wainwright, Nathan W. Bailey.

**Supervision:** J. Benito Wainwright, Graeme D. Ruxton, Nathan W. Bailey.

**Visualization:** J. Benito Wainwright.

**Writing – original draft:** J. Benito Wainwright.

**Writing – review & editing:** J. Benito Wainwright, Charlotte E. J. Rolfe, Graeme D. Ruxton, Nathan W. Bailey.

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
