## [Editor Report · Decision Letter 0]

5 Aug 2025

Dear Dr Wainwright,

Thank you for submitting your revised manuscript entitled "Extraordinary adaptations: Functional and evolutionary synergy of trait components can explain the existence of leaf masquerade" for consideration as a Short Report by PLOS Biology.

Your revisions have now been evaluated by the PLOS Biology editorial staff and I'm writing to let you know that we would like to send your submission out for re-review.

IMPORTANT: I note that you have not supplied a marked-up "track changes" version of your manuscript. Please provide this when you upload your additional metadata (see next paragraph).

However, before we can send your manuscript back to the reviewers, we need you to complete your submission by providing the metadata that is required for full assessment. To this end, please login to Editorial Manager where you will find the paper in the 'Submissions Needing Revisions' folder on your homepage. Please click 'Revise Submission' from the Action Links and complete all additional questions in the submission questionnaire.

Once your full submission is complete, your paper will undergo a series of checks in preparation for peer review. After your manuscript has passed the checks it will be sent out for re-review. To provide the metadata for your submission, please Login to Editorial Manager (https://www.editorialmanager.com/pbiology) within two working days, i.e. by Aug 07 2025 11:59PM.

Kind regards,

Roli Roberts

Roland Roberts, PhD

Senior Editor

PLOS Biology

rroberts@plos.org

---

## [Decision Letter · Decision Letter 1]

3 Oct 2025

Dear Dr Wainwright,

Thank you for your patience while we considered your revised manuscript "Extraordinary adaptations: Functional and evolutionary synergy of trait components can explain the existence of leaf masquerade" for publication as a Short Report at PLOS Biology. This revised version of your manuscript has been evaluated by the PLOS Biology editors, the Academic Editor and the original reviewers.

Based on the reviews, we are likely to accept this manuscript for publication, provided you satisfactorily address the remaining points raised by the reviewers and the following data and other policy-related requests.

IMPORTANT - please attend to the following:

a) We try to avoid punctuation in our Titles. Please change your Title to "Functional and evolutionary synergy of trait components can explain the existence of leaf masquerade in katydids"

b) Please attend to the remaining comments from the reviewers. I asked the Academic Editor about reviewer#3's requests, some of which might involve additional data or analysis; they said, "It to the credit of reviewer #3 to have made such effort to scrutinise the paper. I don't see however any of the suggestions to be critical. These are all manners to verify details. I would give the authors the option to address comments if they already have the data. If they choose not to, they they can fix the edits and move on." I hope that this guidance is helpful.

c) Please address my Data Policy requests below; specifically, we need you to supply the numerical values underlying Figs 1C, 2ABCDE , S1-S5, either as a supplementary data file or as a permanent DOI’d deposition. I note that you already have an associated Zenodo deposition, for which you provide a tiny.url reviewer link. Please could you confirm whether the data and code in this deposition are sufficient to recreate the Figures? If not, please include it in your deposition or as a supplementary file(s).

d) Please cite the location of the data clearly in all relevant main and supplementary Figure legends, e.g. “The data underlying this Figure can be found in S1 Data” or “The data underlying this Figure can be found in https://zenodo.org/records/XXXXXXXX

We expect to receive your revised manuscript within two weeks.

*Published Peer Review History*

*Press*

Sincerely,

Roli Roberts

Roland Roberts, PhD

Senior Editor

rroberts@plos.org

PLOS Biology

DATA POLICY:

Regardless of the method selected, please ensure that you provide the individual numerical values that underlie the summary data displayed in the following figure panels as they are essential for readers to assess your analysis and to reproduce it: Figs 1C, 2ABCDE , S1-S5. NOTE: the numerical data provided should include all replicates AND the way in which the plotted mean and errors were derived (it should not present only the mean/average values).

CODE POLICY

DATA NOT SHOWN?

REVIEWERS' COMMENTS:

Reviewer #1:

I have read the new version of this manuscript and the response of the authors to reviewers, and I should say I am satisfied with the changes and clarifications. I believe that this version has improved content readability, the figures are of good publication standards, therefore I have recommended acceptance.

Reviewer #2:

[identifies himself as Ehab Abouheif]

In this revised version, the Authors have done a good job at responding to most of the critiques I raised from the first round of review. Therefore, I think this manuscript is worthy of publication in PloS Biology. I just have one final comment / critique that I hope that the Authors can address prior to publication, which I believe would clarify the argument the Authors are trying to make.

The Authors propose two alternative hypotheses: "non-adaptive stepwise evolution" versus "Adaptive synergistic evolution" of complex composite traits. However, I remain unclear about what the Authors mean by "Non-adaptive." I believe what the Authors mean is non-adaptive with respect only to leaf masquerading. If so, the Authors should state this clearly. Otherwise, general readers would take these shape and size traits to have no function with respect to the organism, and as such, would evolve neutrally and eventually be lost. I don't think this is what the Authors mean. It is more plausible that these traits serve alternative functions, and where therefore retained by selection for other functions unrelated to leaf-masquerading, and then were co-opted via directional selection to form composite trait with adaptive value for leaf masquerading. The Authors would do well to cite and refer to Vrba and Gould's classic paper on co-opiton in morphological evolution. Basically, what I have described here is co-option from a trait that previously has a different function, what they call "aptation" versus co-option of a trait that previously had no function "non-aptation."

I feel that maintaining and clarifying this distinction between aptation and non-aptation (the Authors don't have to use this exact terminology as it is not frequently used, just describe it) would help readers understand the broad evolutionary significance of the results of this manuscript, and would help readers better understand the what the two alternative scenarios actually mean for the evolution of complex adaptations.

Reviewer #3:

I have carefully assessed the authors' responses to my comments and those of the other reviewers, as well as the overall quality of the revised manuscript.

First, regarding my original concerns, the authors have addressed them comprehensively. For the issue of behavioral experiment design, they clarified that the goal of the experiment was to extract general principles of color-shape interaction rather than mimic specific prey, confirmed through bird-calibrated images that mealworms do not interfere with target classification, and reasonably explained why spectrometry (as suggested in O'Hanlon 2013) was unnecessary due to minimal UV reflection from leaves and bark. On intraspecific variation, they demonstrated that sampled species were monomorphic for measured traits at the study site, used individual-level data and species as a random effect to account for shape variation, and further incorporated sex into analyses—finding no impact on the core evolutionary association between color and shape. For predator vision modeling, they clarified the purpose of human leafiness scoring (to align with avian predation results, not directly substitute avian vision), confirmed minimal UV reflection in katydid tegmina via spectrophotometry, and cited literature to validate humans as proxies for ecologically relevant primates.

These changes have elevated the manuscript from "major revision/rejection" to "minor revision". Two small, easily addressable issues remain:

1. A supplementary control (or at least discussion) eliminating the olfactory/visual confound of the exposed mealworm bait.

2. A brief tetrachromatic visual‐modelling check on a subset of katydid tegmina to confirm that UV contrast is irrelevant (or provide ΔS values if it is).

Both can be handled with additional supplementary material and do not require new field work.

Synthesizing comments from other reviewers and the revised manuscript's quality, there are still areas for improvement. In response to Reviewer 1's suggestion to include Mimetica and Typophyllum, while the authors' explanations (sampling challenges, polymorphism, uncertain phylogeny) are valid, future work could explore expanded sampling and specialized methods for these genera to enhance conclusion generality. For Reviewer 2's concern about phylogenetic path analysis, the authors could present detailed comparisons of alternative models (stepwise, simultaneous, hybrid) in supplementary materials (e.g., tables of path coefficients and fit statistics) to better illustrate why simultaneous evolution is favored. Addressing Reviewer 4's question on "censored data," the authors might add a brief supplementary analysis to show how excluding such data would affect results, reinforcing the robustness of their survival analysis. Additionally, some tables (e.g., Table S3, S4) could have clearer formatting (consistent decimal places, explicit parameter labels) to improve readability.

Overall, the authors have effectively resolved major concerns, and the revised manuscript demonstrates strong scientific rigor—integrating field experiments, phylogenetic analyses, and human perception assays to advance understanding of leaf masquerade evolution. The remaining improvements are minor and do not compromise the core conclusions. I therefore recommend a "Minor Revision" for this manuscript. With these final adjustments, the work will be a valuable contribution to the fields of evolutionary biology and camouflage research.

Reviewer #4:

I think the authors did a good job addressing reviewer concerns, and enhancing their justification for their experimental design. While the methods the authors used are somewhat standard for predation experiments at this point (mealworms are commonly used), the additional clarification is useful for readers who are less familiar with this type of experimental design.

I also appreciate the authors' expanded justification for avoiding predators who have learned to identify leaf mimics, as predators are notoriously good at learning how to find food, and many tropical birds are long lived and highly likely to be experienced with leaf mimics.

Now that the authors have included sex in their analyses of tegmina shape, and have found an effect of sex on aspect ratio, I would like the authors to include sample sizes for sex by species in the methods/supplemental.

Below are some additional comments, line numbers are for the mark up version:

Lines 65 and 68: I disagree with Reviewer 3's insistence on not using the term "leaf-mimicking" here, as insects with wings/tagmina that look like leaves are generally called leaf mimics, and the phenomenon as leaf mimicry (see Wang et al., 2022, Cell, for a recent example).

Line 85: "at sampling" needs to be added here, otherwise the methods aren't clear. "Survival of each individual target at sampling was assessed based on the presence or absence of an edible mealworm bait, which served as the 'body' of the target."

Line 161: I'm not sure I'd agree with the authors that the OU model being the best fit suggests "strong" directional selection. Directional selection, sure, but the word "strong" is pretty vague here. I suggest the authors remove the term "strong" or provide additional justification for their use of the term.

Line 216: Interesting to see that female tegmina have lower aspect ratios compared to males, perhaps the two sexes are mimicking different tree species? And, given that there was an effect of sex on aspect ratio, please be sure to include the sample sizes for sex.

Line 227: please state what the two (or three?) human-perceived metrics of leafiness are here, so readers don't have to remember. If it's all three, then it should be "any" instead of "either".

Line 233: Here and earlier- this statement suggests that leafiness was never lost. Is that correct? That leaf masquerade has been gained multiple times in this system but never lost? Please clarify.

Methods lines 489-494: Please include sample sizes for sex per species.

Lines 242-244: This is a good place to reiterate that females had different aspect ratios than males, further suggesting other selective forces acting on the trait.

Figure 2: The pigment vs non-pigment is much easier to see in the revised figure.

---

## [Editor Report · Decision Letter 2]

16 Oct 2025

Dear Benito,

Thank you for the submission of your revised Short Reports "Functional and evolutionary synergy of trait components can explain the existence of leaf masquerade in katydids" for publication in PLOS Biology. On behalf of my colleagues and the Academic Editor, Abderrahman Khila, I'm pleased to say that we can in principle accept your manuscript for publication, provided you address any remaining formatting and reporting issues. These will be detailed in an email you should receive within 2-3 business days from our colleagues in the journal operations team; no action is required from you until then. Please note that we will not be able to formally accept your manuscript and schedule it for publication until you have completed any requested changes.

Sincerely, 

Roli

Senior Editor

PLOS Biology

rroberts@plos.org